# Differentiation and Identification of Endophytic Bacteria from *Populus* Based on Mass Fingerprints and Gene Sequences

**DOI:** 10.3390/ijms241713449

**Published:** 2023-08-30

**Authors:** Xia Wang, Guanqi Wu, Shuo Han, Jingjing Yang, Xiangwei He, Haifang Li

**Affiliations:** 1College of Biological Science and Technology, Beijing Forestry University, Beijing 100083, China; wx18338911855@163.com (X.W.); qi_w1230@163.com (G.W.); holahan99@bjfu.edu.cn (S.H.); 18813171632@163.com (J.Y.); 2Department of Chemistry, MOE (Ministry of Education) Key Laboratory of Bioorganic Phosphorus Chemistry & Chemical Biology, Tsinghua University, Beijing 100084, China

**Keywords:** endophytic bacteria, pretreatment, MALDI-TOF MS, differentiation, relatedness

## Abstract

Plant endophytic bacteria play important roles in plants’ growth and resistance to stress. It is important to characterize endophytic bacteria to be able to understand their benefits. Matrix-assisted laser desorption ionization–time of flight mass spectrometry (MALDI-TOF MS) has become a powerful technique for bacterial identification due to its high throughput and simple procedures. In this study, the endophytic bacteria separated from *Populus* (the leaves, roots and stems of *Populus tomentosa* Carrière; stems of *Populus nigra* Linn. var. *nigra*; and stems of *Populus canadensis* Moench) were identified and classified based on MALDI-TOF MS data and 16S rRNA gene sequencing. The sampling and preparation of bacteria were optimized to obtain meaningful protein mass fingerprints. The composite correlation index (CCI) values of the inter-genera and inter-species protein mass fingerprints demonstrated sufficient differences between the strains. In the CCI value matrix for ten species in the same genus, all the CCI values were less than 0.5. Among the species, 95.6% of all the CCI values were less than 0.5. After data processing, the classification capacity of the protein mass fingerprints was verified using inter-specific and inter-generic PCoA. To compare different methods’ potential for differentiation and phylogenetic analysis, a dendrogram of the MS profiles and a phylogenetic tree based on the 16S rRNA gene sequences were constructed using 61 endophytic bacteria found in *Populus*. The clustering and grouping results show that the phylogenetic analysis based on MALDI-TOF MS is similar to that based on 16S rRNA gene sequencing. This study provides a valuable reference for differentiating and identifying endophytic bacteria according to their protein mass fingerprints.

## 1. Introduction

Endophytic bacteria live in plant leaves, roots, stems and other tissues [1,2]. These bacteria can directly promote the plant’s growth by regulating or synthesizing plant growth hormones, changing the root’s physiology, promoting nutrient absorption and interacting with the host’s genome to achieve coevolutionary dynamics [3,4,5]. In addition, endophytic bacteria can help plants resist biotic stresses by producing antibiotics or secreting toxins to antagonize crop pathogens and pests [6,7,8]. They can also respond to abiotic stresses, such as drought stress [9]. Furthermore, endophytic bacteria have emerged as potential alternatives to chemical fertilizers and pesticides, contributing to the protection of ecological systems [3,10,11,12]. 

Exploring endophytic bacterial resources requires identifying individual endophytic bacteria, so it is desirable to have different methods for detecting the microbial community within the plant. The initial methods for the classification and identification of bacteria were based on morphological characterization and physiological/biochemical reactions, which were cumbersome, usually took more than 3 days, and required expert skills [13]. These eye-observation methods are prone to producing false-positive results when identifying similar phenotypes [14]. With the rise of molecular biology, microbial classification and identification began to favor DNA sequence analysis in the late 20th century [15]. Since then, sequencing technology based on the polymerase chain reaction (PCR) have been developed for microbial ecology, phylogeny, functional genome analysis and transcriptional profiling [15,16]. If pure strains can be easily obtained, 16S rRNA gene sequencing based on PCR is preferred for bacterial identification due to its high stability and accuracy [13,17,18,19,20]. The preliminary separation and purification of bacteria are required for the high-throughput obtention of information on their content, and a culture-independent DNA sequencing method became appealing for bacterial analysis. This method allows for high-throughput detection even for nonculturable bacteria. Remarkably, bias has reportedly been introduced in almost every step of the various options, which is troubling [21,22].

Matrix-assisted laser desorption–ionization (MALDI) is a soft ionization technique appropriate for the analysis of biomolecules such as nucleic acids, proteins, peptides and lipids [22]. It has become an attractive method for the rapid identification of bacteria [23,24,25], fungi [26,27,28], yeasts [29] and protists [30], even for subspecies-level typing [31]. Claydon et al. first used MALDI-TOF MS to obtain whole-cell spectral fingerprints of *Escherichia coli*, *Staphylococcus saprophyticus*, *Staphylococcus aureus*, *Staphylococcus epidermidis*, *Citrobacter freundii*, *Klebsiella aerogenes* and *Mycobacterium smegmatis* in 1996 [32]. The outstanding advantages of the MALDI-TOF MS-based bacterial identification methods include their rapid and high-throughput preparation of samples on a 384-well target plate, and the detection of a mass spectrum, which can be completed rapidly when a benchtop liquid-handling system is used [33]. MALDI-TOF MS is widely used in clinical settings, and Tsuchida et al. summarized the application of MALDI-TOF MS to clinical bacterial identification [34]. In recent years, some studies have applied MALDI-TOF MS to the analysis of plant microorganisms. Bacterial identification and classification based on MALDI-TOF MS have been used by comparing the similarity of mass spectrometry profiles. For example, a total of 585 potato rhizosphere bacteria were clustered with a similarity threshold of 94.75% [35]. LaMontagne et al. classified 114 bacterial isolates from the endosphere and rhizosphere of maize into 60 coherent MALDI-TOF MS taxonomic units using machine learning methods based on the MS spectra, which were similar to the 16S rRNA gene sequence [36]. Ashfaq et al. reviewed the application of MALDI-TOF MS for identifying environmental bacteria using a bibliometrics analysis method, investigating the characterization of pollutant-degrading, plant-associated, disease-causing and soil-beneficial bacteria, as well as other environmental bacteria [37]. MALDI-TOF MS is a reference-based identification method for microorganisms. Commercial databases such as the VITEK MS system (BioMérieux) and the Bruker Biotyper system (Bruker Daltonics) are often used for microbial identification [36,38]. In Cobo’s research, the identification results for clinically relevant anaerobic bacteria using MALDI-TOF MS and 16S rRNA gene sequencing were compared. The misidentification of five *Phocaeicola* and failure in twelve *Anaerococcus* species when using MALDI-TOF MS was due to the limitations of the Bruker Biotyper database for these rare, infrequent and newly discovered species [39]. However, the outstanding advantages of MALDI-TOF MS -based bacterial identification led the exploration of its applications for diverse microorganisms.

*Populus*, a widely planted tree, hosts diverse species of endophytic bacteria, primarily comprising the phyla Actinobacteria, Proteobacteria, Bacteroidetes and Firmicutes [40]. *Bacillus* and *Pseudomonas* are the dominant genera among the endophytic bacteria, and factors such as high salt stress and soil pH are influential in shaping the composition of endophytic bacterial communities in *Populus* [41,42]. In order to apply the MALDI-TOF MS method to the rapid identification of endophytic bacteria in *Populus*, the differentiation and identification abilities of this method were compared with those of 16S rRNA gene sequences. The MALDI-TOF MS protein fingerprints of endophytic bacteria separated from *Populus* were detected with optimized culture and pretreatment conditions. Simultaneously, the endophytic bacteria were sequenced based on the 16S rRNA gene. The composite correlation index (CCI) values were calculated to judge the relatedness, illustrating this method’s capability for bacterial differentiation based on mass proteins fingerprints. After data processing, the intra-specific and inter-specific bacteria were separated. A dendrogram and a phylogenetic tree for 61 strains of endophytic bacteria, belonging to 22 species, were constructed based on the MS profiles and 16S rRNA gene sequences, respectively. The cluster result of the protein mass profile dendrogram was similar to the 16S rRNA phylogenetic tree based on gene sequences, which shows that MALDI-TOF MS is a viable alternative to 16S rRNA gene sequencing for the differentiation and identification of endophytic bacteria.

## 2. Results

### 2.1. Protein Mass Fingerprints of Endophytic Bacteria

To obtain high-quality protein mass fingerprints of endophytic strains, the growth periods, culture media, sample pretreatment methods and incubation temperature were optimized. As shown in Figure 1, with *Pseudomonas koreensis* as a model, MS profiles obtained under different conditions were compared. The strongest peaks of the strains in the exponential growth period were very different from those in the stationary period and death period, and the number of peaks was also relatively low (Figure 1A). This demonstrated that the protein expression levels changed with the growth period. There were slight differences in the protein mass fingerprints with the NA, LB and R_2_A media. The number of characteristic peaks of endophytic bacteria was the highest when the bacteria were cultured in the LB medium, especially for those with a molecular weight greater than 8 kDa (Figure 1B). As shown in Figure 1C, the influence of incubation temperature was reflected in the MS profiles. Compared with 25 °C, the temperature of 30 °C was more suitable for endophytic bacterial growth, so an MS profile with more protein peaks was obtained. At the higher temperature of 37 °C, there was almost no peak above 8000 (*m/z*), indicating that high temperature has a negative effect on endophytic bacteria. Figure 1D shows that the MS profiles from the third (extracted protein) method were missing many targeted peaks compared with those obtained with the lysed cell and whole-cell detection methods. This suggests that the pretreatment process led to the loss of some proteins. According to the above results, in our experiments, the optimal methods were determined to be LB medium and 30 °C for culture, with sampling in the stationary period, and using the second (detected lysed cell) pretreatment method. The reproducibility and stability of the optimized methods were verified using *Pseudomonas koreensis* (Appendix A). 

### 2.2. Composite Correlation Index (CCI) Matrix for Protein Mass Fingerprints

The protein mass fingerprints at the genus, species and strain levels and inter-genera and inter-species CCI matrix were combined to estimate the potential of MALDI-TOF MS in the systematic classification of bacteria. At the strain level, the MS profiles were almost the same and the differences among seven MS profiles were in the relative intensity of the peaks (Figure 2A). At the species level, for ten species of *Pseudomonas*, four groups of peptide/protein peaks all appearing in ten MS profiles were marked, and the peaks simultaneously appearing in more than one MS profiles were also observed (Figure 2B). At the genus level, comparing *Bacillus*, *Rhizobium*, *Pseudomonas* and *Microbacterium* showed that almost none of the same peptide/protein peaks appeared simultaneously in four MS profiles, although a few peaks appeared in two MS profiles (Figure 2C). In the CCI matrix, more peaks at the same position in two MS profiles and the higher CCI value indicate the closer relationship of the two bacteria. The CCI value of 1.0 in blue indicates a high correlation, and 0 in white indicates a large difference between the two strains’ MS profiles. In the inter-generic CCI matrix, all of the CCI values were less than 0.1 (Figure 3A), demonstrating an enormous difference in cellular content at a genus level. In the inter-specific CCI matrix, the distribution of the CCI values was as follows: 18 groups had values less than 0.1, accounting for 20%; 68 groups had values between 0.1 and 0.5, accounting for 75.6%; and 4 groups had values greater than 0.5, accounting for 4.4% (Figure 3B). These differences provided a basis for the classification of species. 

### 2.3. PCoA to Verify the Differentiation Capability of MALDI-TOF MS 

The differentiation and identification capabilities of MALDI-TOF MS for endophytic strains were verified. Principal coordinates analysis (PCoA) based on Bray–Curtis dissimilarities was performed for bacterial classification. The PCoA of the raw data from the biological replicates of three strains (*Pseudomonas psychrotolerans* YN25D, *P. baetica* AN9D and *P. graminis* BN23D) did not reveal clear strain separations (Appendix A). The data processing method was optimized based on the clustering heatmap. Correct data processing should group the biological replicates of the strains together and produce a regular heatmap. As a control, the raw data clustering heatmap is shown in Appendix A; the heatmap is irregular, and the differentiation of strains is confused. Regularity did not appear, even after the raw data were cleaned with a de-noised measure (Appendix A). To eliminate huge differences and retain slight differences among the different protein mass fingerprints, log10 conversion was performed on the peak intensity. In the de-noising and log10 conversion data clustering heatmap, regularity was revealed, and the strains of *P. baetica* AN9D, *P. graminis* BN23D and *P. psychrotolerans* YN25D were distinctly separated (Appendix A). As shown in Figure 4A, the PCoA of the fifteen biological replicates of *P. baetica* AN9D, *P. graminis* BN23D and *P. psychrotolerans* YN25D demonstrated a clear strain separation. To verify the differentiation for genera, 43 endophytic bacteria belonging to *Bacillus*, *Microbacterium* and *Pseudomonas* were selected and analyzed (Figure 4B). The clear grouping observed at both the species and genus levels demonstrated the high differentiation capacity of MALDI-TOF MS when identifying endophytic bacteria.

### 2.4. Evaluation of Relatedness Based on MS Profiles

In order to assess the relatedness among species, a dendrogram based on the MALDI-TOF MS protein mass profiles and a 16S rRNA gene phylogenetic tree for 61 strains of endophytic bacteria belonging to 22 species (Appendix A) were constructed (Figure 5 and Figure 6). In both the dendrogram and the phylogenetic tree, 61 strains of endophytic bacteria were divided into four clusters. The genus clusters were *Pseudomonas*, *Microbacterium*, *Rhizobium* and *Bacillus*. There was a slight difference between the dendrogram and the phylogenetic tree. In the dendrogram, *Bacillus* was adjacent to the branch including *Pseudomonas* and *Microbacterium*, and *Rhizobium*, with five endophytic bacteria strains, was distant from the other genera. In the phylogenetic tree, the genus *Rhizobium* adhered to the cluster of *Pseudomonas*, and the genera *Bacillus* and *Microbacterium* were located in a large branch together. At the species level, the grouping similarity was 90.9% between the dendrogram and the phylogenetic tree. A total of 20 out of 22 species had consistent grouping, with the exceptions being *B. cereus* and *B. tropicus*. The Sankey diagram (Figure 7) shows the obvious grouping relationship between the MS dendrogram and the phylogenetic tree. Both methods grouped the genera *Microbacterium*, *Bacillus* and *Rhizobium* together. *Pseudomonas*, including forty strains, belonged to seven groups. All the species were separated into one group, except for *P. reidholzensis* and *P. putida*, *P. poae* and *P. canadensis*, which were grouped together.

## 3. Discussion

MALDI-TOF MS has emerged as a powerful tool for the identification of bacteria, and the construction of commercial databases has rapidly developed, especially for human pathogens [39]. Databases for environmental bacteria are limited [36,38]. In this study, MALDI-TOF MS was applied to classify and analyze the evolutionary relationships of endophytic bacteria based on differential protein expression.

### 3.1. The Effects of Bacterial Pretreatment and Culture Conditions on MS Profiles

In this study, protein mass fingerprints were obtained after optimizing the pretreatment and culture conditions. The culture conditions, including the growth period, culture medium and incubation temperature, can influence the presence and intensity of bacterial protein expression [43,44]. The growth period can lead to the differential expression of ribosomal proteins. During the exponential growth period, small ribosomal proteins are expressed to support rapid growth and division. In the stationary growth period, larger ribosomal proteins are expressed to maintain metabolic activity. In the death period, protein synthesis slows down or stops. As shown in Figure 1A, *Pseudomonas koreensis* presented different protein expression levels in different periods. In the exponential growth period, proteins beyond the 8 kDa mass range were scarce, and the most abundant proteins were distributed in the 3–4 kDa and 7–8 kDa mass ranges. In the death period, the diversity in proteins expression decreased, which led to fewer protein peaks. During the stationary phase, both the diversity and intensity of protein expression were higher. More proteins were detected, especially proteins with molecular weights greater than 8 kDa. Hence, the stationary period is most suitable for sampling.

In this study, compared with those obtained with NA and R_2_A media, the endophytic bacteria mass fingerprints obtained from the LB medium exhibited the greatest abundance of peaks (Figure 1B). R_2_A and NA are low-nutrient culture media commonly used for selecting and screening bacteria, which can influence bacterial protein expression. In contrast, LB is a nutrient-rich complete medium suitable for rapid bacterial growth, which promotes the comprehensive expression of the proteins.

The influence of the incubation temperature on protein expression was also taken into account. Temperature can affect the growth period and expression of ribosomal proteins in bacteria. Additionally, the composition and abundance of ribosomal proteins also responded to changes in temperature [45]. It was observed that endophytic bacterial growth was optimal at 30 °C (Figure 1C), while a high temperature had a deleterious impact on growth.

The simplest pretreatment method is to directly detect whole bacterial cells, which is suitable for Gram-negative bacteria, but not for Gram-positive bacteria with thick cell walls [39]. After protein extraction using formic acid/acetonitrile, the noise was reduced, but some targeted proteins were lost. The reason for this was that the centrifugation step led to some proteins being retained in the precipitate (Figure 1D). By contrast, the lysed cell detection method could release intracellular proteins and remove small-molecule noise while retaining the signals of macromolecular proteins. Therefore, the lysed cell method is more suitable for endophytic bacteria pretreatment.

### 3.2. Bacterial Classification Based on MALDI-TOF MS

Prior studies have reported that MALDI-TOF MS can be used to detect conserved ribosomal proteins [46]. MALDI-TOF MS has found wide-ranging applications in microbiology, including (but not limited to) the discovery of new species [47], bacterial typing [48] and species identification [29]. The success of MALDI-TOF MS in microbiology can be attributed to its robust recognition and identification capabilities. Under consistent detection conditions, the difference in the MS profiles of seven strains of *Pseudomonas koreensis* is mainly reflected in the relative peak intensity (Figure 2A). Bacterial proteins are highly conserved, meaning that the protein composition of a particular species of bacteria remained mostly the same under the same growth conditions. However, slight variations in the intensity of protein peaks can reflect changes in the abundance of the protein expression of the bacteria. The sensitivity of MALDI-TOF MS is demonstrated by its ability to detect even these small differences in protein composition.

Compared with the MS profiles of inter-generic endophytic bacteria, the similar characteristics of bacteria protein mass fingerprints within one genus increased. The MS profiles of ten bacteria belonging to *Pseudomonas* had four of the same groups of peptide/proteins peaks, and the number of peaks in two or more strains increased significantly (Figure 2B). These same peaks contributed to the classification of the species into one genus in the PCoA (Figure 4B). The identification of distinct protein peaks using MALDI-TOF MS facilitated the discrimination of different species within a genus based on their unique protein mass fingerprints. As shown in Figure 3B, the differences in the MS profiles of bacteria within a genus were quantified according to the CCI values, of which 75.6% exceeded 0.1 and only 4.4% exceeded 0.5. The MS profiles of four strains of endophytic bacteria belonging to the genera *Bacillus*, *Rhizobium*, *Pseudomonas* and *Microbacterium* showed that almost none of the same peptide/protein peaks appeared simultaneously (Figure 2C), and the CCI values among four genera were all less than 0.1 (Figure 3A). The MS profiles, CCI values and bacterial relatedness are interconnected. A lower CCI value indicated a greater dissimilarity between protein mass fingerprints, and this dissimilarity was the basis for the identification of strains. As the classification level decreased, the MS profiles became more similar, resulting in a larger mean value for the CCI matrix, which could aid in determining the relatedness of strains. However, the majority of CCI values were less than 0.5, providing sufficient differences for the identification of different strains, and the PCoA depicted in Figure 4 supports this conclusion. 

It should be mentioned that it is difficult to distinguish variations from the original MS profiles. With the aid of optimized data processing methods, including logarithmic conversion and noise reduction, clear and accurate separation can be achieved not only at the strain level but also at the genus level. Noise reduction is used to remove high-intensity noise, and logarithmic conversion is used to narrow the range of variables and retain small differences in intensity. 

### 3.3. Protein Mass Fingerprints vs. 16S rRNA Gene Sequence

A phylogenetic tree based on the 16S rRNA gene sequences is generally used to evaluate the relatedness among bacteria. In this study, we explored the construction of a dendrogram based on protein mass fingerprints with 61 endophytic bacteria belonging to 22 species. Comparing the dendrogram and phylogenetic tree (Figure 5 and Figure 6) showed that both methods can effectively distinguish *Rhizobium*, *Microbacterium*, *Bacillus* and *Pseudomonas*. It could be seen that the cluster in the dendrogram was similar to that in the phylogenetic tree to some extent. The reason for this is that the phylogenetic tree was based on the sequences of the 16S rRNA gene, while the MS profiles were based on proteins expressed by genes. Both proteins and genes can be used to accurately determine the genetic relationships of species due to their conserved protein and variables. The peaks observed in the protein mass fingerprints can provide classification results that are consistent with the 16S rRNA gene sequences. The slight differences in the clade were that the species of *B. cereus* and *B. tropicus* were mixed together in the dendrogram. The reason for this could be ascribed to the distinct topological structures between the two methods. In general, different methods for the judgment of relatedness will lead to slightly different interpretations or evaluations of the positions of small clades, possibly due to different computational strategies [49,50]. A Sankey diagram was used to compare the relationships between the dendrogram and phylogenetic trees (Figure 7). The grouping was based on the distance in the MS profile dendrogram and the frequency in the 16S rRNA gene phylogenetic tree. Consistent results indicate that MALDI-TOF MS could be used as an alternative to the 16 rRNA gene sequences in terms of resolution.

## 4. Methods and Materials

### 4.1. Populus’s Endophytic Bacterial Strains

Three species of *Populus* (*Populus tomentosa* Carrière*; Populus nigra* Linn. var. *nigra*; and *Populus canadensis* Moench) were chosen for this experiment. Plants were grown in Shandong, China (115.45° E, 35.47° N). The root and stem tissues, with a length of 10 mm, width of 10 mm and thickness of 3–5 mm, were taken from 1.5 m above and below the ground, and leaf tissues at high altitude with natural thickness were taken as samples. The ages of the *Populus* specimens were 1–2, 5–6 and 9–10 years. For *Populus tomentosa* Carrière, samples of different tissues and ages were collected, while for *Populus nigra* Linn. var. *nigra* and *Populus canadensis* Moench, only stems that were 5–6 years old were collected. Further details are shown in Appendix A.

Surface sterilization procedures were strictly followed and involved rinsing with 75% ethanol for 30 s, soaking in 8% sodium hypochlorite for 3 min and rinsing five times with sterile water. A 20 μL volume of the last rinse solution was dripped and spread on LB agar plates, and the *Populus* tissues were rolled on the LB agar plates. This procedure was conducted to assess the plant surface’s sterilization [51]. After surface sterilization, the tissues were mashed, suspended in PBS buffer solution and diluted with sterile water to different concentration gradients (10^−1^–10^−6^). Subsequently, the tissue dilution solution was spread onto separate isolation media, including nutrient agar (NA) (beef extract 5.0 g, pancreatic digest of casein 10.0 g, sodium chloride 5.0 g, agar 17.0 g and distilled water 1 L) and R_2_A agar (casein hydrolysate 0.5 g, yeast extract 0.5 g, pancreatic digest of casein 0.5 g, glucose 0.5 g, soluble starch 0.5 g, sodium acetate 0.3 g, magnesium sulfate 0.05 g, potassium dihydrogen phosphate 0.3 g, agar 15.0 g and distilled water 1 L). The plates were incubated at 30 °C for 1–3 d. Independent colonies were purified three times using the streak plate method, preserved in 30% glycerin and stored in a −80 °C freezer. 

### 4.2. MALDI-TOF MS Sample Preparation and Detection

The culture media, incubation temperature and growth periods were optimized using a single-factor experiment. The three types of culture media used were R_2_A, NA and Luria-Bertani medium (tryptone 10 g, yeast extract 5 g, sodium chloride 10 g and distilled water 1 L). The incubation temperature was set at 25, 30 and 37 °C, respectively. Samples were taken at 4, 12 and 18 h, corresponding to the exponential growth period, stationary period and death period, respectively. All culture was carried out in liquid medium under rotation at 150 rpm. In addition, three pretreatment methods were employed to prepare endophytic bacteria with the optimized culture methods for detection. The first method was the direct detection of whole bacterial cells: suspending the cells after the endophytic bacterial cells were washed three times with deionized H_2_O and mixing the cells with CHCA solution (1:1, *v*/*v*). The second method was the detection of cell lysis: inactivating bacterial cells with 75% alcohol after washing, adding an appropriate amount of 70% formic acid and acetonitrile in a 1:1 mixture directly and adding CHCA solution (1:1, *v*/*v*). The third method was the detection of the extracted proteins, which was similar to the second method, except that the proteins were extracted using 70% formic acid/acetonitrile and centrifuged at 12,000 rpm. The optimized culture and pretreatment method were used to process the endophytic bacteria isolated from *Populous*.

Endophytic bacterial cells were detected on a MALDI-TOF/TOF MS instrument with a 337 nm laser (AXIMA Performance, Shimadzu, Japan). Briefly, 20 mg of α-cyano-4-hydroxycinnamic acid (CHCA) (Sigma, St. Louis, MO, USA) was dissolved in 1 mL of acetonitrile/H_2_O (1:1, *v*/*v*) and used as a matrix solution. For MALDI-TOF MS detection, 1.0 μL of sample and 1.0 μL of matrix mixture were mixed and spotted on a stainless plate. Using a random pattern for laser shots, 200 profiles were collected for each mass spectrum. The endophytic bacterium *Pseudomonas koreensis* CGMCC 1.15630 (China General Microbiological Culture Collection Center, CGMCC) was used to optimize the culture and pretreatment methods. Calibration was carried out with freshly cultured *Escherichia coli* ATCC 8739 (American Type Culture Collection) grown in LB agar for 12–18 h. Detection was carried out in the 2–20 kDa *m/z* range in positive linear mode. 

### 4.3. Mass Spectra Data Processing

The raw mass spectra data were exported from the Shimadzu Biotech MALDI-MS software 2.9.3; the *m/z* was accurate to one bit, and the signal intensity of peaks was averaged to within one bit. The logarithms with base 10 of the peak intensities of each protein’s mass fingerprints were taken. Noise reduction was performed by selecting 100 intensities of continuous *m/z*, in which the first 20% of the peak intensities were chosen as the signal peaks, while the remaining peaks were considered noise and assigned values of 0. Each unit was repeated by moving 1 Da and iterating from 2 to 20 kDa. These processes are implemented through C sharp.

### 4.4. 16S rRNA Gene Sequence

Endophyte isolates were identified simultaneously with the sequencing of 16S rRNA genes. Briefly, the 16S rRNA gene was extracted from pure cultured endophytic bacteria via thermal lysis and amplified using RuiBio BioTech primers 1492R (5′-GGTTACCTTGTTACGACTT-3′) and 27F (5′-AGAGTTTGATCCTGGCTCAG-3′) (Beijing Ruibo Biological Co., Ltd, Beijing, China). The PCR thermal profile consisted of an initial denaturation at 94 °C for 2 min, followed by 32 cycles of denaturation at 94 °C for 30 s, annealing at 56 °C for 30 s, extension at 72 °C for 90 s and a final extension step at 72 °C for 5 min, within a total reaction volume of 25 µL. 16S rRNA gene PCR products were purified and sequenced by Beijing Ruibo Biological Co., Ltd. (Beijing, China).

### 4.5. Statistical Methods

Composite correlation index (CCI) matrix heatmaps based on the Pearson correlation coefficient were utilized to evaluate the relatedness among the MS profiles at the genus and species levels. This analysis was implemented using the *stats* R package. Principal coordinates analysis (PCoA) based on Bray–Curtis dissimilarities was employed to assess the classification ability of MALDI-TOF MS. This analysis was carried out using the *vegan* and *ade4* R packages. Additionally, a mass profile dendrogram was constructed using the Biozeron Cloud Platform (http://www.cloud.biomicroclass.com/CloudPlatform, accessed on 18 September 2022). This dendrogram was generated using the single-cluster method and included 61 strains of endophytic bacteria. This dendrogram can be used to compare the potential of protein mass fingerprints for phylogenetic analysis with that of the 16S rRNA gene sequence. The amplified 16S rRNA sequences were aligned using the BLAST database (https://blast.ncbi.nlm.nih.gov/Blast.cgi, accessed on 22 November 2021). A phylogenetic tree for the 61 endophytic strains was constructed by combining the resulting 16S rRNA gene sequences using the maximum composite likelihood model calculated through the neighbor-joining method with 1000 bootstrap replicates in MEGA version 11 software.

## 5. Conclusions

In conclusion, this study elucidated the excellent performance of protein mass fingerprints for endophytic bacterial differentiation and phylogenetic systematics. The endophytic bacteria from *Populus* were detected using MALDI-TOF MS with optimized culture and pretreatment methods. The inter-genera and inter-species relatedness were assessed using CCI values. All the inter-generic CCI values were less than 0.1, indicating significant differences in distinguishing endophytic bacteria. CCI values greater than 0.1 accounted for 80% in the inter-specific matrix, indicating that the correlation between two inter-specific strains was closer than that between inter-generic strains. CCI values greater than 0.5 accounted for only 4.4%, providing sufficient differences for the differentiation species within the same genus. PCoA was used to classify the inter-specific and inter-generic strains with processed data. Clear and correct differentiation demonstrated a credible classification ability. An MS profile dendrogram for 61 strains of endophytic bacteria was constructed using protein mass fingerprints to evaluate the systematic classification capability of MALDI-TOF MS. Compared with phylogenetic tree based on 16 rRNA gene sequences, the similarity of the results shows the excellent performance of protein mass fingerprints in systematic classification analysis. Considering its high throughput and high sensitivity for proteins, MALDI-TOF MS is a promising alternative to 16S rRNA gene sequencing for the identification of endophytic bacteria and has great potential for bacterial typing tracing and the classification of new species.

## Figures and Tables

**Figure 1 ijms-24-13449-f001:**
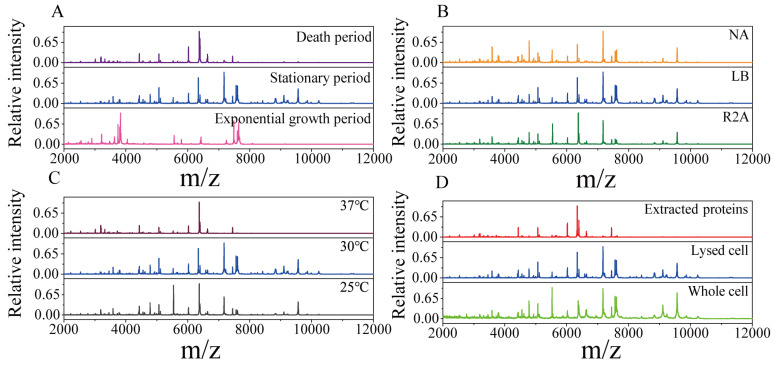
Protein mass fingerprints for the endophytic bacteria *Pseudomonas koreensis* under different culture or pretreatment conditions. (**A**) Three sampled growth periods; (**B**) cultured with three media; (**C**) incubated at three temperatures and (**D**) the three pretreatment methods used.

**Figure 2 ijms-24-13449-f002:**
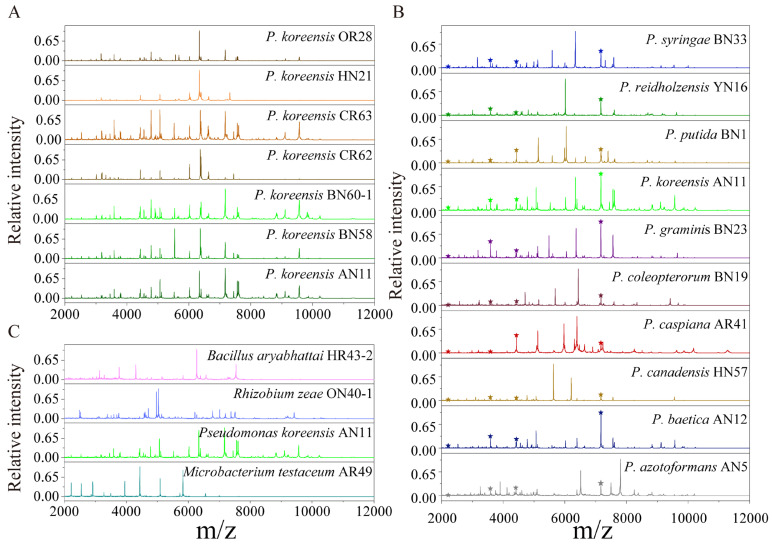
The comparison of MS profiles at the strain, species and genus levels. (**A**) Strain levels of *P. koreensis* OR28, *P. koreensis* HN21, *P. koreensis* CR63, *Pseudomonas koreensis* CR62, *P. koreensis* BN60, *P. koreensis* BN58 and *P. koreensis* AN11; and (**B**) species level: *P. azotoformans* AN5, *P. baetica* AN12, *P. canadensis* HN57, *P. caspiana* AR41, *P. coleopterorum* BN19, *P. graminis* BN23, *P. koreensis* AN11, *P. putida* BN1, *P. reidholzensis* YN16 and *P. syringae* BN33. ★ means that the peak appeared in all the MS profiles; (**C**) Genera levels: *Bacillus aryabhattai* HR43, *Rhizobium zeae* ON40, *Pseudomonas koreensis* AN11 and *Microbacterium testaceum* AR49.

**Figure 3 ijms-24-13449-f003:**
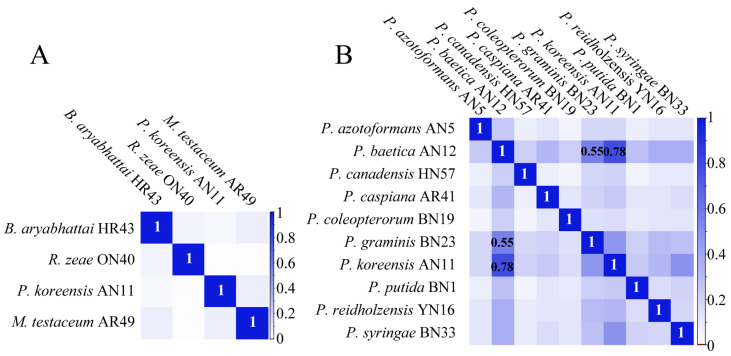
CCI matrix derived from different strains belonging to (**A**) *Bacillus aryabhattai*, *Rhizobium zeae*, *Pseudomonas koreensis* and *Microbacterium testaceum*; and (**B**) *Pseudomonas azotoformans* AN5, *P. baetica* AN12, *P. canadensis* HN57, *P. caspiana* AR41, *P. coleopterorum* BN19, *P. graminis* BN23, *P. koreensis* AN11, *P. putida* BN1, *P. reidholzensis* YN16 and *P. syringae* BN33.

**Figure 4 ijms-24-13449-f004:**
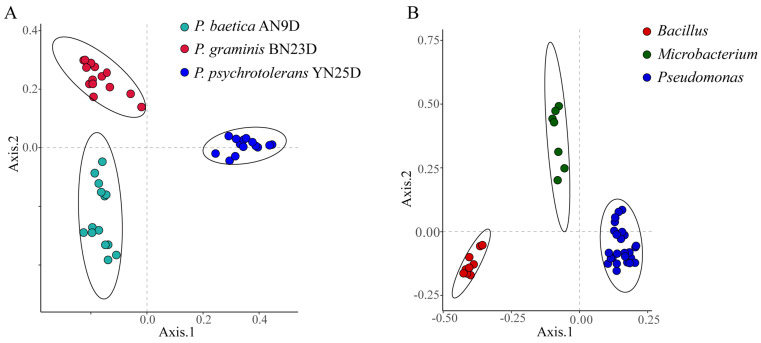
Principal coordinates analysis (PCoA) based on Bray-Curtis dissimilarities demonstrating the distance between MALDI-TOF MS profiles. (**A**) Fifteen biological replicates of each of *P. baetica* AN9D, *P. graminis* BN23D and *P. psychrotolerans* YN25D. (**B**) Forty-three species’ endophytic bacteria, belonging to *Bacillus*, *Microbacterium* and *Pseudomonas*.

**Figure 5 ijms-24-13449-f005:**
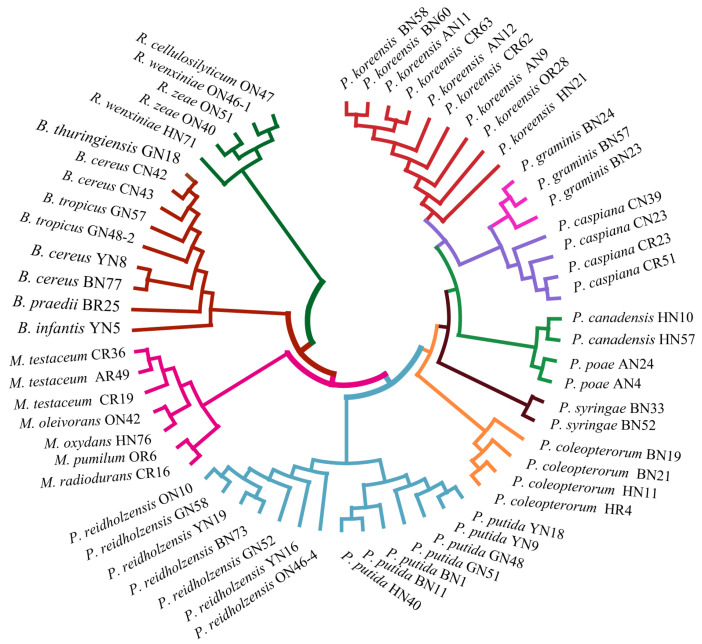
Phylogenetic tree based on MALDI-TOF MS profiles of 61 endophytic bacteria.

**Figure 6 ijms-24-13449-f006:**
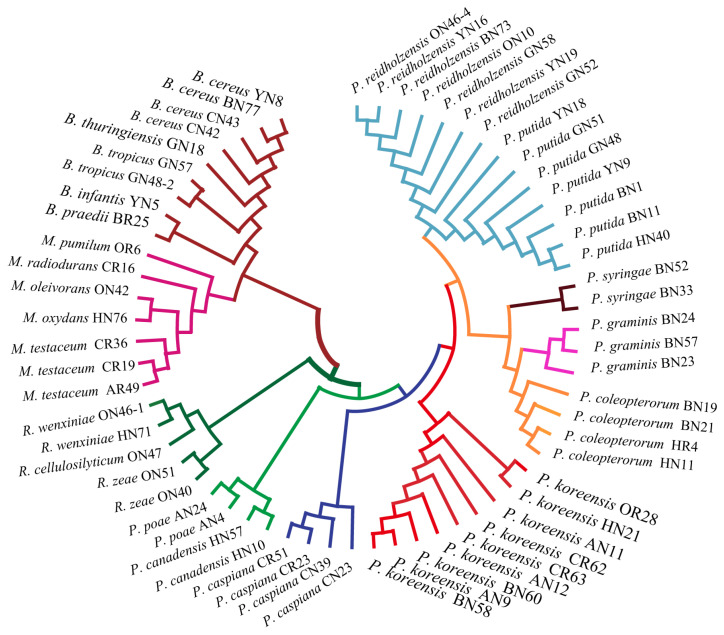
Phylogenetic tree based on the 16S rRNA gene sequence of 61 endophytic bacteria.

**Figure 7 ijms-24-13449-f007:**
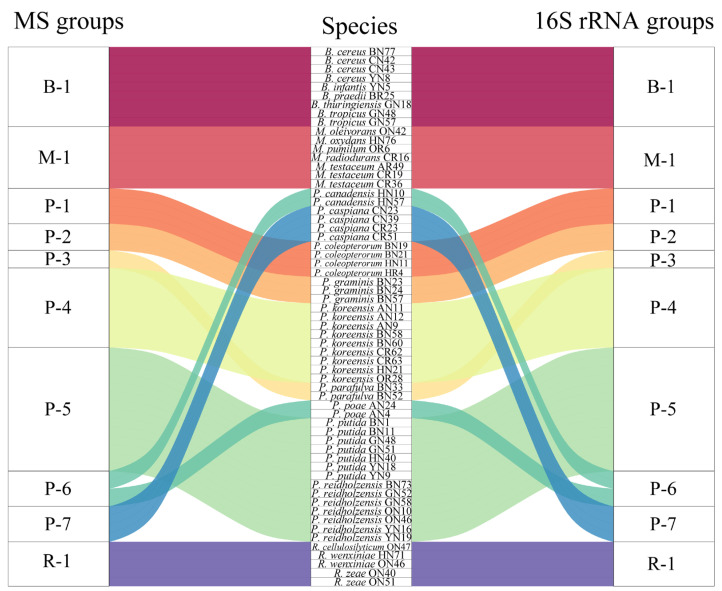
Sankey diagram based on the relationship between the MALDI-TOF MS profile and the 16S rRNA gene sequence.

## Data Availability

The data used to support the findings of this study are available from the corresponding author upon request.

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
