# Peer review of "Differentiation and Identification of Endophytic Bacteria from Populus Based on Mass Fingerprints and Gene Sequences"

_ijms, 2023, doi:10.3390/ijms241713449_

Round 1

Reviewer 1 Report (New Reviewer)

In the introduction the manuscript should mention " doi: 10.1016/j.anaerobe.2023.102754." and comment if the authors found a similar drawback with bacteria from populus.

Author Response

Point 1: In the introduction the manuscript should mention " doi: 10.1016/j.anaerobe.2023.102754." and comment if the authors found a similar drawback with bacteria from populus.

Response 1: The MALDI-TOF MS-based bacterial identification has outstanding advantages of simple preparation requirement, high-throughput and rapid speed. Its ability and accuracy of identification of diverse microorganisms were concerned.

In Cobo’s research, the accuracy of identification results by MALDI-TOF MS and 16S rRNA gene sequencing was compared. It was found that the absence of the corresponding microorganisms in the used Bruker’s database led to the failure and misidentification. In the commercial database like Bruker Biotyper system, clinical microorganisms were dominant and plant bacteria were poorly involved. In our research, the differentiation and identification of bacteria from Populus based on MALDI-TOF MS was compared to gene sequences to evaluate the reliability of MALDI-TOF MS in plant endophytic bacteria 

The corresponding citation of “In Cobo's research, the identification results of clinically relevant anaerobic bacteria by MALDI-TOF MS and 16S rRNA gene sequencing were compared. The misidentification of five Phocaeicola and failure in twelve Anaerococcus species by MALDI-TOF MS was due to the limitations of the Bruker Byotyper database for these rare, infrequent and newly discovered species.” was supplemented in the introduction. It was cited as 39 references (lines 81-92).

Reviewer 2 Report (New Reviewer)

The manuscript entitled "Differentiation and Identification of Endophytic Bacteria from Populus Based on Mass Fingerprints and Gene Sequences" is interesting and has the potential to interest readers. The issues presented in this manuscript are consistent with the topics of the "International Journal of Molecular Sciences". The "Introduction" chapter provides sufficient background and relevant references, and the study design is appropriate. Also the results have been well described, and the conclusions correspond to the research goal set by the authors.

Manuscript requires a some of improvements before publication in the Journal “International Journal of Molecular Sciences”.

1.      In the "Methods and Materials" chapter, separate a separate subchapter in which you will briefly describe the statistical methods used in your research. This will streamline the "Methods and Materials" chapter.

2.      Please, be sure that all the references cited in the manuscript are also included in the reference list and vice versa with matching spellings and dates.

3.      In the "References" section for all references, add doi. This will make it easier to find the publication.

Author Response

Point 1: In the "Methods and Materials" chapter, separate a separate subchapter in which you will briefly describe the statistical methods used in your research. This will streamline the "Methods and Materials" chapter.

Response 1: The section on statistical methods has been made into a separate section (lines 186-208)

Point 2: Please, be sure that all the references cited in the manuscript are also included in the reference list and vice versa with matching spellings and dates.

Response 2: All of the references have been checked and cited in the manuscript.

Point 3: In the "References" section for all references, add doi. This will make it easier to find the publication.

Response 3: The doi of all references has been added to the References section.

Reviewer 3 Report (New Reviewer)

The presented work with the use of modern methods of analysis and identification of microorganisms was thought to be original and relevant, but only at first glance.

In fact, the presence of novelty is not clear. After all, databases already exist for the identification of microorganisms by the MALDI-TOF MS method. Don't endophytic microorganisms fit in there?

Why are these three species of Populus taken, what is the difference between their endophytic microorganisms?

Why were the bacteria of the four genera represented examined?

On fig. 1 conditions for pseudomonads, but for bacilli, the optimal conditions will be different (both temperature, and cultivation time, and, possibly, nutrient medium). There is no versatility.

How will the system you propose behave in a real assessment of the diversity of endophytes (not just four genera)? Try to predict, announce.

Or maybe it is better to use metagenomic sequencing to solve the problems posed in this work?

Author Response

Point 1: The presented work with the use of modern methods of analysis and identification of microorganisms was thought to be original and relevant, but only at first glance.

In fact, the presence of novelty is not clear. After all, databases already exist for the identification of microorganisms by the MALDI-TOF MS method. Don't endophytic microorganisms fit in there?

Response 1: Although MALDI-TOF MS has been employed in some microbial identification, in commercial databases clinical microorganisms were dominant and plant bacteria were poorly involved. More than 40% Populus bacteria used in our research was not included in the commercial database. So this work tried to verify the application of MALDI-MS to plant endophytic bacteria. 

Point 2: Why are these three species of Populus taken, what is the difference between their endophytic microorganisms?

Response 2: These three Populus species are most widely planted in China. The endophytic bacterial communities of the three Populus were significantly different, the diversity was Populus nigra Linn. var. nigra > Populus canadensis Moench > Populus tomentosa Carrière.

Point 3: Why were the bacteria of the four genera represented examined?

Response 3: More than 100 species of endophytic bacteria were separated from the three Populus species. Most of endophytic bacterial species with the same genus was less than 2. To enhance the universality of the research, we selected strains within genera that encompassed four or more species for the relevant research work.

Point 4: On fig. 1 conditions for pseudomonads, but for bacilli, the optimal conditions will be different (both temperature, and cultivation time, and, possibly, nutrient medium). There is no versatility.

Response 4: Endophytic bacteria derived from the Populus were isolated and cultivated under uniform conditions for detecting. While diverse bacterial species exhibit distinct optimal growth requirements, our objective in establishing protein mass fingerprints does not involve detecting them under their individual optimum growth conditions. Rather, our aim is to identify a growth condition that is relatively advantageous and applicable to the majority of bacterial species. Consequently, we chose to optimize culture conditions through the utilization of Pseudomonas koreensis, which was present in all Populus in this study.

Point 5: How will the system you propose behave in a real assessment of the diversity of endophytes (not just four genera)? Try to predict, announce.

Or maybe it is better to use metagenomic sequencing to solve the problems posed in this work?

Response 5: The main purpose of this study is to evaluate the accurate classification and identification by MALDI-TOF MS for endophytic bacteria. Of course, we agree with you that the metagenomic sequencing is a good choice for assessment of the diversity of endophytes, we can use it in our future work.

Round 2

Reviewer 3 Report (New Reviewer)

The authors answered most of the questions and eliminated the comments.

But it is not entirely clear why the optimal conditions for Pseudomonas koreensis are considered to be optimal for all endophytic microorganisms? But what about bacteria of other genera (besides the four presented in the article) and species?

Author Response

Point 1: The authors answered most of the questions and eliminated the comments.

But it is not entirely clear why the optimal conditions for Pseudomonas koreensis are considered to be optimal for all endophytic microorganisms?

Response 1: The aim of this study is to compare the capabilities of 16S rRNA gene sequence and MALDI-TOF MS fingerprints in the classification of endophytic bacteria. To ensure getting high-quality mass protein fingerprints of endophytic bacterial, it is important to establish universal cultivation condition for most of Populus endophytic bacteria.

Pseudomonas is a pivotal microbial group. In our research on Populus endophytic bacterial culturomics, we found that 20% of the isolated endophytic bacteria from Populus belong to the Pseudomonas, which is dominant among the endophytic bacteria. Therefore, the optimal conditions for Pseudomonas are discussed. The conditions for other bacteria are also considered. Regarding the cultivation temperature, 30°C is suitable and commonly used for endophytic bacteria cultivation. Considering the potential impact of varied composition of different culture media on sample analysis, LB medium was chosen as the culture medium due to its broad applicability to all isolated endophytic bacteria. In terms of sampling time, the stationary period of each endophytic bacteria was selected for sampling and detection.

The results of MALDI-TOF MS detection demonstrated that mass protein fingerprints of all endophytic bacteria obtained under these culture conditions are good enough to support the congruent outcomes between the phylogenetic analysis and 16S rRNA gene sequence analysis.

Point 2:But what about bacteria of other genera (besides the four presented in the article) and species?

Response 2: In our research on plant endophytic bacterial culturomics, a total of 112 species belonging to 43 genera were isolated. Prominent genera among them include Pseudomonas, Rhizobium, Microbacterium, and Bacillus. The primary objective of our study is to assess the capabilities of MALDI-TOF MS fingerprint profiles and 16S rRNA sequences in the classification of plant endophytic bacteria.

The results derived from MALDI-TOF MS fingerprints have demonstrated that the variation between mass protein fingerprints of different genera is significantly greater than that observed between species. This conspicuous difference allows for a relatively straightforward differentiation during classification analysis. Consequently, in selecting the strains for this study, we deliberately opted for endophytic bacteria exhibiting multi-level classification attributes. This approach has enabled the establishment of a multidimensional phylogenetic tree that facilitates diverse analyses.

This manuscript is a resubmission of an earlier submission. The following is a list of the peer review reports and author responses from that submission.

Round 1

Reviewer 1 Report

The article «Isolates and Differentiation of Endophytic Bacteria in Plants Based on Proteins Mass Fingerprinting '' is devoted to the important and acute theme of investigation of endophytic bacterial communities. Authors demonstrate the possibility of using MALDI MS for endophytes identification. I don't question the identical results of MS and PCoA analysis, but now the main idea of the article is “MALDI-TOF MS were expected to reach the level of gene systematics based on 16S rRNA gene sequences. This study provides valuable application to isolate and differentiate endophytic bacteria by their protein mass fingerprinting”, but authors didn’t obtain results which evidenced this sentence.

According to the literature, “ Bacterial identification by MS is based on matching with the MALDI mass spectrum of the species (strains) listed in the database. If the species is not listed in the database or is a new species, such an organism will not be identified and instead must be classified by another method. One of the pitfalls of bacterial identification is that identification by mass spectrometry involves a matching of the mass spectra of the bacteria (strains) listed in the database; therefore, bacteria that are not listed in the database cannot be identified…. Current recommendations suggest that users follow maintenance, inspection, and accuracy control methods (e.g., calibration using E. coli) defined by each company; the measurement of standard strains is essential for accurate control of clinical microbiology tests “Tsuchida, S.; Nakayama, T. MALDI-Based Mass Spectrometry in Clinical Testing: Focus on Bacterial Identification. Appl. Sci. 2022, 12, 2814. https://doi.org/10.3390/app12062814

But now in the area of wide range plant endophytes there are not any standard strains, and possibly, will not. 

“MALDI-TOF-MS is useful for the rapid identification of Gram-negative rods, but not Gram-positive bacteria” Bishop, B.; Geffen, Y.; Plaut, A.; Kassis, O.; Bitterman, R.; Paul, M.; Neuberger, A. Clin The use of matrix-assisted laser desorption/ionization time-of-flight mass spectrometry for rapid bacterial identification in patients with smear-positive bacterial meningitis. Microbiol. Infect. 2018, 24, 171–174. [Google Scholar] [CrossRef][Green Version]

Thus, MALDI-TOF-MS in clinics is used for differentiation of pathotype (for example) of more or less well-known bacteria, characteristic for human (human pathogenesis) and now it cannot be used for endophytic communities investigation without genome sequence.

The article needs major changes. Authors must be focused on the Populus endophytic community, investigated with two methods, not on MALDI-TOF-MS.

Now the endophytic community of Populus isn’t discussed in this paper. Please, insert discussion on this topic and conclusions on composition of endophytic bacteria in Populus. 

I have some comments:

1)Title: It should be rewritten: “Identification of Endophytic Bacteria in Populus (species of populus under investigation!) Plants Based on Proteins Mass Fingerprinting and DNA sequencing”

2)Abstract: should be rewritten according to context. Species of populus must be inserted.

3)Introduction: Please, insert previous data on endophytic communities of other trees or populus. Replace lines 83-92 with description of the aim of your investigation (investigation of POpulus internal microflora)

4)Materials and methods:

Give species, variety and place of populus collection. What protocol of sterilization was used?

Results

Conditions of bacteria cultivation before MALDI-TOF-MS were given only for Gram- Pseudomonas (Fig 1). Authors must provide information on Gram+ bacteria cultivation and preparation, since Gram+ investigation with this method is doubtful.

Discussion

Lines 271-280 Is it literature data? 

Lines 293-296 - repetition

255-258 - Authors say that MALDI-TOF-MS is not useful for Gram+, but they used it and showed the result in this article. Please, explain.

Minor editing of English language required.

Author Response

Point 1: The article «Isolates and Differentiation of Endophytic Bacteria in Plants Based on Proteins Mass Fingerprinting '' is devoted to the important and acute theme of investigation of endophytic bacterial communities. Authors demonstrate the possibility of using MALDI MS for endophytes identification. I don't question the identical results of MS and PCoA analysis, but now the main idea of the article is “MALDI-TOF MS were expected to reach the level of gene systematics based on 16S rRNA gene sequences. This study provides valuable application to isolate and differentiate endophytic bacteria by their protein mass fingerprinting”, but authors didn’t obtain results which evidenced this sentence.

Response 1: Thank you for your comments. The similarity of MS dendrogram to 16S rRNA gene phylogenetic tree was evaluated as supplementary results in lines of 285-287 and 297-299 in the text:

At the species level, the grouping similarity was 90.9% between the MS dendrogram and the phylogenetic tree. A total of 20 out of 22 species had consistent grouping, with the exceptions being B. cereus and B. ropicus.

Point 2: According to the literature, “Bacterial identification by MS is based on matching with the MALDI mass spectrum of the species (strains) listed in the database. If the species is not listed in the database or is a new species, such an organism will not be identified and instead must be classified by another method. One of the pitfalls of bacterial identification is that identification by mass spectrometry involves a matching of the mass spectra of the bacteria (strains) listed in the database; therefore, bacteria that are not listed in the database cannot be identified…. Current recommendations suggest that users follow maintenance, inspection, and accuracy control methods (e.g., calibration using E. coli) defined by each company; the measurement of standard strains is essential for accurate control of clinical microbiology tests “Tsuchida, S.; Nakayama, T. MALDI-Based Mass Spectrometry in Clinical Testing: Focus on Bacterial Identification. Appl. Sci. 2022, 12, 2814. https://doi.org/10.3390/app12062814

But now in the area of wide range plant endophytes there are not any standard strains, and possibly, will not.

Response 2: We agree with you that the limitation of using MALDI-MS to identification of plant endophytic bacteria. MALDI-TOF MS has been commonly employed for identification of clinical microorganisms by commercial database. At present, the proportion of plant endophytic bacteria included in the commercial databases is relatively low. So it is significant to develop MALDI-MS based analysis methods for plant endophytic bacteria.

The calibration is an accuracy control used to ensure the reproducibility of MS spectra. In our experiments, standard strain of E. coli is also used to make a superior calibration for obtaining reproducible MS spectra.

Point 3: “MALDI-TOF-MS is useful for the rapid identification of Gram-negative rods, but not Gram-positive bacteria” Bishop, B.; Geffen, Y.; Plaut, A.; Kassis, O.; Bitterman, R.; Paul, M.; Neuberger, A. Clin The use of matrix-assisted laser desorption/ionization time-of-flight mass spectrometry for rapid bacterial identification in patients with smear-positive bacterial meningitis. Microbiol. Infect. 2018, 24, 171–174. [Google Scholar] [CrossRef][Green Version]。

Thus, MALDI-TOF-MS in clinics is used for differentiation of pathotype (for example) of more or less well-known bacteria, characteristic for human (human pathogenesis) and now it cannot be used for endophytic communities investigation without genome sequence.

Response 3: Compared with Gram-negative, Gram-positive bacteria has thicker cell walls. Gram-positive bacteria MS detection needs to break the cell walls firstly. Many studies have used MALDI-TOF MS for Gran-positive bacteria identification. For examples, Romero-Gomez et al. achieved an identification accuracy of 97.84% for Gram-positive bacteria in blood cultures [1]; Schulthess et al. attained a 95.6% accuracy for identifying Gram-positive cocci[2]; TeKippe et al. optimized the identification methods for clinically relevant Gram-Positive organisms with MALDI-TOF MS [3].

In our study, Gram-positive bacteria was lysed with acetonitrile containing 70% formic acid, and abundant protein fingerprints in MS spectra could be gotten.

References

[1]. Romero-Gomez, M.P; Gomez-Gil, R.; Pano-Pardo, J.R.; Mingorance, J. Identification and susceptibility testing of microorganism by direct inoculation from positive blood culture bottles by combining MALDI-TOF and Vitek-2 Compact is rapid and effective. J Infect 2012, 65, 513-520).

[2]. Schulthess, B.; Brodner, K.; Bloemberg, G.V.; Zbinden, R.; Bttger, E.C.; Hombach, M. Identification of Gram-Positive cocci by use of matrix-assisted laser desorption ionization–time of flight mass spectrometry: comparison of different preparation methods and implementation of a practical algorithm for routine diagnostics. J Clin Microbiol. 2013, 51, 1834-1840).

[3]. TeKippe, E.M.; Shuey, S.; Winkler, D.W.; Butler, M.A.; Burnham, C.A.D. Optimizing identification of clinically relevant Gram-Positive organisms by use of the Bruker Biotyper matrix-assisted laser desorption ionization–time of flight mass spectrometry system. J Clin Microbiol. 2013, 51, 1421-1427.

Point 4: Thus, MALDI-TOF-MS in clinics is used for differentiation of pathotype (for example) of more or less well-known bacteria, characteristic for human (human pathogenesis) and now it cannot be used for endophytic communities investigation without genome sequence.

The article needs major changes. Authors must be focused on the Populus endophytic community, investigated with two methods, not on MALDI-TOF-MS.

Now the endophytic community of Populus isn’t discussed in this paper. Please, insert discussion on this topic and conclusions on composition of endophytic bacteria in Populus.

Response 4: In our study, both MALDI-TOF MS detection and 16S rRNA sequencing have been taken for each plant isolates. The two methods results were presented in Abstract, Results and Discussion section in the revised manuscript.

The diversity of Populus endophytic community has been freshly presented in the introduction section (lines 103-108). In our study, the selected 61 strains were only a small percentage of the whole endophytic bacteria community of Populus, so it is less important to discuss the community.

Point 5: Title: It should be rewritten: “Identification of Endophytic Bacteria in Populus (species of populus under investigation!) Plants Based on Proteins Mass Fingerprinting and DNA sequencing”

Response 5: Thank you for your suggestion. In our study, we not only evaluate the identification ability of MALDI-TOF MS and 16S rRNA gene sequencing, but also discuss the differentiation ability. So the title is changed to “Differentiation and Identification of Endophytic Bacteria from Populus Based on Proteins Mass Fingerprints and 16S rRNA Gene Sequences”.

Point 6:  Abstract: should be rewritten according to context. Species of populus must be inserted.

Response 6:  The species of Populus (Populus tomentosa Carrière, Populus nigra Linn. var. nigra and Populus canadensis Moench) were inserted in Abstract (lines 19-20).

Point 7:  Introduction: Please, insert previous data on endophytic communities of other trees or populus. Replace lines 83-92 with description of the aim of your investigation (investigation of Populus internal microflora)

Response 7: We have inserted the Populus’ endophytic communities reported in previous data (lines 103-108).

Point 8:  Materials and methods:

Give species, variety and place of populus collection. What protocol of sterilization was used?

Response 8: Species, variety and place of populus collection are list in the Supplementary Table 1. The protocol of sterilization was described in detail in the Materials and methods section (lines 125-127).

Point 9: Results

Conditions of bacteria cultivation before MALDI-TOF-MS were given only for Gram- Pseudomonas (Fig 1). Authors must provide information on Gram+ bacteria cultivation and preparation, since Gram+ investigation with this method is doubtful.

Response 9: In our study, the finally used cell lysis pretreatment method was also suitable for Gram-positive bacteria: inactivating bacterial cells with 75% alcohol after washing, adding an appropriate amount of formic acid and acetonitrile directly, and mixing the mixture with CHCA solution for MALDI-MS detection. The proteins mass fingerprints of the Gram-positive stains of Microbacterium testaceum AR49, CR36 and CR19 were added in Figure S2 in Supplementary Material.

Point 10:  Lines 271-280 Is it literature data? Lines 293-296 - repetition.

Response 10:  Lines 271-280 before modification is not literature data.

Original sentence: Growth period can lead to differential expression of ribosomal proteins. During exponential growth period, small ribosomal proteins are expressed to support rapid growth and division. In stationary growth period, larger ribosomal proteins expressed to maintain metabolic activity. In death period, protein synthesis slows down or stops. As shown in Figure 1C, Pseudomonas koreensis presented different protein expression levels at different periods. In the exponential growth period, proteins beyond the 8kDa mass range are scare and the most abundant proteins distributed in 3-4kDa and 7-8kDa mass ranges. In the death period, the proteins expression diversity decreases, which leads to fewer protein peaks. During the stationary phase, both protein expression abundance and diversity are higher. More proteins were detected, especially for the proteins with molecular weight beyond 8kDa. Hence, the stationary period is suitable for sampling.)

The duplicate has been deleted.

Original sentence: Under consistent detection conditions, differences in the MS profiles of seven strains of Pseudomonas koreensis are mainly in relative peak intensities (Figure 2A).

Point 11:  255-258 - Authors say that MALDI-TOF-MS is not useful for Gram+, but they used it and showed the result in this article. Please, explain.

Response 11: The original sentence of "The simplest pretreatment method is directly detecting whole bacterial cells, which is suitable for Gram-negative bacteria but not for Gram-positive bacteria with thick cell walls."

It is not mean that MALDI-TOF MS is not applicable to Gram-positive bacteria, its meaning is the simplest pretreatment method is not suitable for Gram-positive bacteria due to the thick cell walls. The cell lysis preparation used in the experiments could get good Gram-positive bacteria MS spectra.

Reviewer 2 Report

the work uses MS analysis to characterize the microbial content of poplar

problems:

requires editing  spaces/spellings  and many sentences where key words seem to be missing  =  leaves you guessing

methods are inadquate    

the results do not seem to correspond to the methods     in terms of medium  source of isolates

the work focusses on pseudomonad bacillus and microbacterium isolates but i did not pick up that these were from the plant  nor what organ in the plant they were found   may be the total plant was assayed

the method used would not distinguish epiphytes from endophytes  think that the work endophytes should not be part of the title 

would expect to see separate replicates for one strain under different conditions   do not see that 

the finding that there are different peaks between isolates is not unexpected 

needs extensive professional and scientific editing

sentences often have missing words     usual spellings etc 

but often use of words is confusing    i really do not agree with their use of the term isolation          to me as a microbiologist it is not identification  

Author Response

Point 1: requires editing spaces/spellings and many sentences where key words seem to be missing = leaves you guessing

Response 1: We first performed a comprehensive review and revision of the manuscript,then the manuscript was professionally revised by editing service provided by MDPI.

Point 2: methods are inadquate

the results do not seem to correspond to the methods in terms of medium source of isolates.

the work focusses on pseudomonad bacillus and microbacterium isolates but i did not pick up that these were from the plant nor what organ in the plant they were found may be the total plant was assayed

Response 2: Table 1 in Supplementary material provides a list of the sources of analyzed bacterial strains, and the Introduction section added relevant information about the endophytic bacterial community. The endophytic bacterial community associated with Populus is primarily composed of the phyla Actinobacteria, Proteobacteria, Bacteroidetes, and Firmicutes, Bacillus and Pseudomonas are the dominant genera among the endophytic bacteria. (Lines 103-108).

Point 3: how old was the plant? why populus where was it grown and how?

Response 3: The Populus ages, grown place and isolated tissue are list in Table 1 in Supplementary material.

Point 4: the method used would not distinguish epiphytes from endophytes think that the work endophytes should not be part of the title.

Response 4: Before isolating endophytic bacteria from Populus tissues, surface sterilization of the tissues was necessary to remove the epiphytes. The sterilization process was added to the manuscript (lines 125-127).

Point 5: would expect to see separate replicates for one strain under different conditions do not see that the finding that there are different peaks between isolates is not unexpected.

the finding that there are different peaks between isolates is not unexpected.

Response 5: The reproducibility with optimized culture and pretreatment methods under 10 batches was list in the Figure S1 in Supplementary material. There were minimal differences observed in the protein mass fingerprints between different batches.

Point 6: The question in document peer-review-297112950.v1.pdf was list as follows:

Point 6.1: The missing words, missing space and inappropriate words.

but often use of words is confusing, i really do not agree with their use of the term isolation to me as a microbiologist it is not identification.

Response 6.1: We have thoroughly reviewed the entire manuscript and made the necessary changes. The use of “isolation” has also been strictly checked.

Point 6.2: Semantic ambiguity, such as: “In genera CCI matrix, all CCI”, “exchanging genetic information with hosts to regulate”, “MALDI-TOF MS based de-replication screening of potato rhizosphere microorganisms has been reported”, et,al.

Response 6.2: The relevant questions have been resolved. Such as:

  1. “In genera CCI matrix, all CCI” changed to “In CCI values matrix of ten species belong to the same genus”;
  2. “exchanging genetic information with hosts to regulate” changed to “interacting with the hosts genome to achieve coevolutionary dynamics”.
  3. “MALDI-TOF MS based de-replication screening of potato rhizosphere microorganisms has been reported” changed to “The bacteria identification and classification based on MALDI-TOF MS were used by comparing the similarity of mass spectrometry profiles. For example, a total of 585 potato rhizosphere bacteria were clustered with a similarity threshold of 94.75%”.

Point 6.3: how old was the plant why populus where was it grown and how

lab field etc. how were the tissues cleaned how do you insure these are endophytes and not epiphytes etc.

this method as described would also give epiphytes

and yes an endophyte may also grow as an epiphyte should be discussed.

Response 6.3: Species, age and place of Populus tissue collection were list in the Table 1 in Supplementary. To remove the epiphytes, the surface sterilization was performed before the isolation of endophytic bacteria step. The detail was added in Materials and methods.

Point 6.4: why 30 C? seems high. what is NA R2A must have details?

Response 6.4: For the plant endophytic bacteria, many study applied 30 °C to culture the bacteria. For example, Visioli’ group culture Ni hyperaccumulator Noccaea caerulescens at the temperature of 30 °C [1]. We also found that the proteins mass fingerprints quality was better and bacterial growth was also faster at a cultivation temperature of 30 °C. Therefore, we selected 30° C as the optimal cultivation temperature.

Detailed information of NA and R2A has been added to Methods and Materials (Lines 132-136).

[1]. Visioli, G; D'Egidio, S; Vamerali, T; Mattarozzi, M; Sanangelantoni, A.M. Culturable endophytic bacteria enhance Ni translocation in the hyperaccumulator Noccaea caerulescens. Chemosphere 2014 117, 538-544.

Point 6.5: how were the cells grown for the examination

what about sporulators were there spores too.

Response 6.5: In our experiments, the endophytic bacterial cells were cultured in LB medium and incubated at a temperature of 30°C. To account for the potential influence of spores, we conducted sampling at the beginning of the stable period.

Point 6.6: E coli related issues.

what medium to what growth age

you state E coli above put into methods.

where are E coli results, these could be close to Rhizobiun.

Response 6.6: The calibration strain of E.coli was grown in LB agar for 12-18h, and experimental issues of E.coli has been added to methods section 2.2 on line 162. E.coli was used to calibrate the instrument. It is not endophytic bacteria in plants; therefore, it was not used during the cultivation and pretreatment condition optimization.

Point 6.7: where did this bacterium come from? (Pseudomonas koreensis)

Response 6.7: The Pseudomonas koreensis CGMCC 1.15630 was purchased from China General Microbiological Culture Collection Center(CGMCC).

Point 6.8: unclear are all these cultures from the poplar? (Figure 2)

Response 6.8: All these cultures in Figure 2 are isolates from poplar. Table 1 in Supplementary material provides a list of the sources of analyzed bacterial strains in this manuscript.

Point 6.9: Think you should provide say scans of the results from three independent extracts from one bacterium

Response 6.9: In our study, the collection of endophytic bacterial protein mass fingerprints was carried out by the optimized method with three repetitions.

Round 2

Reviewer 1 Report

I have received more or less satisfactory answers on my comments.

Author Response

Point 1: I have received more or less satisfactory answers on my comments.

Response 1: Thank you for all the comments given to make our research more perfect.

Reviewer 2 Report

The paper still requires edits 

and still the methods are inadequate as are legends  for figures  

The paper still has problems with understanding some sentences

Choice of words problem in some cases 

Many sentences could be shortened without loss of sense  etc  

Author Response

Response to Reviewer 2 Comments

Point 1: The comments directly pointed out in the manuscripts.

Response 1: Complete modification were performed in our manuscript.

Point 2: is it not the peptide fragments not the actual proteins that are seen rephrase. Mentioned on page 1, lines 20-21.

how do you know you only have proteins in your extracts need verification. Mentioned on page 10, lines 295-296.

Response 2: Intracellular proteins can be hydrolyzed under strong acid or heating conditions, but our pretreatment methods, using 70% formic acid/acetonitrile at room temperature for pretreatment, only lysed the cytomembrane of endophytic bacteria, so the proteins cannot be hydrolyzed. Previous investigations demonstrated that the peaks detected by MALDI-TOF MS come from intracellular proteins of bacteria (references of 13 and 26 in the manuscript).

Point 3: are you attempting to say it is based on culture morphology unclear. Mentioned on page 2, lines 47-48.

Response 3: Identification methods based on morphological and physiological/biochemical reactions generally requires more than 3 days to obtain identification results. This ambiguous statement has been revised in the Introduction section (line 48).

Point 4: wording does not seem accurate

You can use the rRNA sequences without culturing so think this biasis not correct. Mentioned on page 2, lines 54-57.

Response 4: The inaccurate expression of molecular biology-related methods has been corrected in the Introduction section (lines 51-52, 56-60).

Point 5: you need description of plants age etc. Mentioned on page 3, lines 113-114.

Response 5: The species, age (1-2, 5-6, 9-10 years old), sampling tissues and sampling location of Populus spp. are listed in Table 1 in Supplementary Material. (Table 1 is also attached at the end of this file.)

Point 6: what was expected product size? Mentioned on page 4, lines 159-160.

Response 6: The PCR products of 16S rRNA gene used in this study were 1466 bp.

Point 7: log stationary death is the normal progression

need to give times in culture etc. Mentioned on page 4, lines 173-174.

Response 7: The culturing time of exponential growth period, stationary period and death period are 5, 15 and 25 h, respectively, which was detailed in the Methods and Materials section (lines 135-138).

Point 8: but where are the actual data

just words is OK for a talk not a paper. Mentioned on page 4, lines 185-186.

Response 8: The raw data of the protein mass fingerprints in the mass range of 2-20 kDa is added to the Supplementary Material. In the previous report, Strejcek et.al found that mass weight of many ribosomal proteins is distributed in 4-10 kDa, and in this range the abundance of unique peaks is also the highest (reference of 38 in manuscript).

Point 9: do not understand differences lyzed and between extracts whole cells your methods are incomple. Mentioned on page 5, lines 194-195.

Response 9: The details of the three pretreatment methods were introduced in the Methods and Materials section (lines 150-159). Compared with the lysed cell and whole cell detection methods, some substances were dropped into the precipitation during the centrifugation of the extracted proteins detection method, which made the abundance of the protein mass fingerprints peaks low.

Point 10: what medium for C what conditions of growth

there is inadequate descriptions of how this was done in methods

what growth conditions again need methods and better legends. Mentioned on page 10, lines 218-219.

Response 10: The culture and pretreatment methods were optimized by single factor experiment. The medium for C was LB. The relevant introduction has been added to the Methods and Materials section (lines 135-141, 150-152).

Point 11: how did you avoid having contamination from proteins from culture medium

these likely coat materials. Mentioned on page 10, lines 305-306.

Response 11: To avoid the contamination of proteins from culture media, we firstly washed the bacteria cells three times using deionized H2O and then further treatment was performed.

Point 12 not enough details in methods to understand the verification of these data. Mentioned on page 10, lines 319-320.

Response 12: The details about the methods of culture and pretreatment methods have been added to the Methods and Materials section (lines 135-141, 150-152).

Point 13: none of this is discussed in enough detail

the accepted rules for publishing had to be that the whole study could be repeated from what is in the publication@ could not do with your work. Mentioned on page 11, lines 362-363.

Response 13: Data processing included logarithmic transformation of peak intensity and noise reduction of protein mass fingerprints, which has been added to the Methods and Materials section (lines 170-175).

Table 1. The strains and the isolation source (for Point 5)

Strains

Source

Collection place

Microbacterium testaceum AR49

5-6 years old, the stem of Populus tomentosa Carrière

Shandong province, China

Pseudomonas koreensis AN12

5-6 years old, the stem of Populus tomentosa Carrière

Shandong province, China

Pseudomonas koreensis AN9

5-6 years old, the stem of Populus tomentosa Carrière

Shandong province, China

Pseudomonas koreensis AN11

5-6 years old, the stem of Populus tomentosa Carrière

Shandong province, China

Pseudomonas poae AN24

5-6 years old, the stem of Populus tomentosa Carrière

Shandong province, China

Pseudomonas poae AN4

5-6 years old, the stem of Populus tomentosa Carrière

Shandong province, China

Bacillus cereus BN77

1-2 years old, the stem of Populus tomentosa Carrière

Shandong province, China

Bacillus praedii BR25

1-2 years old, the stem of Populus tomentosa Carrière

Shandong province, China

Pseudomonas coleopterorum BN19

1-2 years old, the stem of Populus tomentosa Carrière

Shandong province, China

Pseudomonas coleopterorum BN21

1-2 years old, the stem of Populus tomentosa Carrière

Shandong province, China

Pseudomonas graminis BN23

1-2 years old, the stem of Populus tomentosa Carrière

Shandong province, China

Pseudomonas graminis BN24

1-2 years old, the stem of Populus tomentosa Carrière

Shandong province, China

Pseudomonas graminis BN57

1-2 years old, the stem of Populus tomentosa Carrière

Shandong province, China

Pseudomonas parafulva BN33

1-2 years old, the stem of Populus tomentosa Carrière

Shandong province, China

Pseudomonas parafulva BN52

1-2 years old, the stem of Populus tomentosa Carrière

Shandong province, China

Pseudomonas koreensis BN58

1-2 years old, the stem of Populus tomentosa Carrière

Shandong province, China

Pseudomonas koreensis BN60

1-2 years old, the stem of Populus tomentosa Carrière

Shandong province, China

Pseudomonas putida BN1

1-2 years old, the stem of Populus tomentosa Carrière

Shandong province, China

Pseudomonas putida BN11

1-2 years old, the stem of Populus tomentosa Carrière

Shandong province, China

Pseudomonas reidholzensis BN73

1-2 years old, the stem of Populus tomentosa Carrière

Shandong province, China

Bacillus cereus CN42

9-10 years old, the stem of Populus tomentosa Carrière

Shandong province, China

Bacillus cereus CN43

9-10 years old, the stem of Populus tomentosa Carrière

Shandong province, China

Microbacterium radiodurans CR16

9-10 years old, the stem of Populus tomentosa Carrière

Shandong province, China

Microbacterium testaceum CR19

9-10 years old, the stem of Populus tomentosa Carrière

Shandong province, China

Microbacterium testaceum CR36

9-10 years old, the stem of Populus tomentosa Carrière

Shandong province, China

Pseudomonas koreensis CR62

9-10 years old, the stem of Populus tomentosa Carrière

Shandong province, China

Pseudomonas koreensis CR63

9-10 years old, the stem of Populus tomentosa Carrière

Shandong province, China

Pseudomonas caspiana CN23

9-10 years old, the stem of Populus tomentosa Carrière

Shandong province, China

Pseudomonas caspiana CN39

9-10 years old, the stem of Populus tomentosa Carrière

Shandong province, China

Pseudomonas caspiana CR23

9-10 years old, the stem of Populus tomentosa Carrière

Shandong province, China

Pseudomonas caspiana CR51

9-10 years old, the stem of Populus tomentosa Carrière

Shandong province, China

Bacillus thuringiensis GN18

5-6 years old, the root of Populus tomentosa Carrière

Shandong province, China

Bacillus tropicus GN48-2

5-6 years old, the root of Populus tomentosa Carrière

Shandong province, China

Bacillus tropicus GN57

5-6 years old, the root of Populus tomentosa Carrière

Shandong province, China

Pseudomonas putida GN48

5-6 years old, the root of Populus tomentosa Carrière

Shandong province, China

Pseudomonas putida GN51

5-6 years old, the root of Populus tomentosa Carrière

Shandong province, China

Pseudomonas reidholzensis GN52

5-6 years old, the root of Populus tomentosa Carrière

Shandong province, China

Pseudomonas reidholzensis GN58

5-6 years old, the root of Populus tomentosa Carrière

Shandong province, China

Microbacterium oxydans HN76

5-6 years old, the stem of Populus nigra Linn. var. nigra

Shandong province, China

Pseudomonas coleopterorum HN11

5-6 years old, the stem of Populus nigra Linn. var. nigra

Shandong province, China

Pseudomonas coleopterorum HR4

5-6 years old, the stem of Populus nigra Linn. var. nigra

Shandong province, China

Pseudomonas koreensis HN21

5-6 years old, the stem of Populus nigra Linn. var. nigra

Shandong province, China

Pseudomonas putida HN40

5-6 years old, the stem of Populus nigra Linn. var. nigra

Shandong province, China

Pseudomonas canadensis HN10

5-6 years old, the stem of Populus nigra Linn. var. nigra

Shandong province, China

Pseudomonas canadensis HN57

5-6 years old, the stem of Populus nigra Linn. var. nigra

Shandong province, China

Rhizobium wenxiniae HN71

5-6 years old, the stem of Populus nigra Linn. var. nigra

Shandong province, China

Microbacterium oleivorans ON42

5-6 years old, the stem of Populus canadensis Moench

Shandong province, China

Microbacterium pumilum OR6

5-6 years old, the stem of Populus canadensis Moench

Shandong province, China

Pseudomonas koreensis OR28

5-6 years old, the stem of Populus canadensis Moench

Shandong province, China

Pseudomonas reidholzensis ON10

5-6 years old, the stem of Populus canadensis Moench

Shandong province, China

Pseudomonas reidholzensis ON46-4

5-6 years old, the stem of Populus canadensis Moench

Shandong province, China

Rhizobium cellulosilyticum ON47

5-6 years old, the stem of Populus canadensis Moench

Shandong province, China

Rhizobium wenxiniae ON46-1

5-6 years old, the stem of Populus canadensis Moench

Shandong province, China

Rhizobium zeae ON40

5-6 years old, the stem of Populus canadensis Moench

Shandong province, China

Rhizobium zeae ON51

5-6 years old, the stem of Populus canadensis Moench

Shandong province, China

Bacillus cereus YN8

5-6 years old, the leaf of Populus tomentosa Carrière

Shandong province, China

Pseudomonas putida YN18

5-6 years old, the leaf of Populus tomentosa Carrière

Shandong province, China

Pseudomonas putida YN9

5-6 years old, the leaf of Populus tomentosa Carrière

Shandong province, China

Pseudomonas reidholzensis YN16

5-6 years old, the leaf of Populus tomentosa Carrière

Shandong province, China

Pseudomonas reidholzensis YN19

5-6 years old, the leaf of Populus tomentosa Carrière

Shandong province, China

Bacillus infantis YN5

5-6 years old, the leaf of Populus tomentosa Carrière

Shandong province, China

Round 3

Reviewer 2 Report

the rewritten manuscript contains so many grammatical problems

You need professional help in getting to a scientific review

Terrible     waste of a reviewers efforts  so much is poorly worded